# Phenotypic Diversity of *Lactobacillus casei* Group Isolates as a Selection Criterion for Use as Secondary Adjunct Starters

**DOI:** 10.3390/microorganisms8010128

**Published:** 2020-01-17

**Authors:** Alessia Levante, Elena Bancalari, Martina Tambassi, Camilla Lazzi, Erasmo Neviani, Monica Gatti

**Affiliations:** Department of Food and Drug, University of Parma, 43124 Parma, Italy; alessia.levante@unipr.it (A.L.); martina.tambassi@gmail.com (M.T.); camilla.lazzi@unipr.it (C.L.); erasmo.neviani@unipr.it (E.N.); monica.gatti@unipr.it (M.G.)

**Keywords:** *L.**casei* group, adjunct starters, impedance analysis, acetoin, gene expression

## Abstract

Autochthonous lactic acid bacteria (LAB) play a key role in the development of cheese flavor. As the pasteurization treatment on raw milk causes the elimination of LAB, secondary starter cultures are used in cheese manufacture to obtain cheeses with improved and standardized flavors. In this work, strains of the *L. casei* group isolated from traditional Italian cheeses were screened for their phenotypic features of technological interest for use as secondary starters. Their milk acidifying performance and the production of volatile compounds when grown in milk were evaluated. Simultaneously, the acetoin metabolic pathway presence was screened in the strains and assessed for its transcriptional activation. The results showed that the analyzed strains, despite belonging to taxonomically-related species, vary greatly according to the measured phenotypes. Four strains among the fourteen screened could be potentially used as adjunct cultures for cheese-making processes. The strain that showed the highest production of acetoin upregulated the aspartate pathway. An increased knowledge of volatile compounds’ production and acidifying properties of LAB strains isolated from traditional dairy products might guide the selection of strains for industrial applications.

## 1. Introduction

Before the introduction of starter cultures in the nineteenth Century, the microbiota of cheese was mainly composed of adventitious microorganisms from milk and the environment. As a consequence, the quality of the cheese was influenced from the inconsistency of the composition of the microbiota. Today, consumers demand cheeses of consistent quality, without defects, but also with more intense flavors [1,2,3].

In this context, the use of starter cultures in cheese manufacture has the objective of obtaining cheeses with improved and standardized quality. Conversely, the use of commercial dairy starter cultures may level out the differences between cheeses, because the few starter lactic acid bacteria species in use are not able to fully represent the diversity of cheeses microbiota [4,5].

Traditional raw-milk cheeses are highly appreciated for their flavors, as opposed to industrial-scale products, which are considered standardized [6]. This is a direct consequence of the pasteurization treatments which lead to the elimination of the microbiota, mainly composed by autochthonous lactic acid bacteria (LAB), that has a key role in flavor development. For this reason, the food industry has performed intensive screening programs to select LAB cultures able to improve cheese flavor [7,8]. 

Over time, dairy companies have changed the composition of mixed starter cultures, introducing adjunct cultures of mesophilic lactobacilli selected for their ability to produce volatile compounds with more intense and distinctive flavor notes [9].

Adjunct cultures are defined as those added to cheese for purposes other than lactic acid production [10]. Indeed, compared to primary starters with acidifying function (starter lactic acid bacteria, SLAB), non-starter lactic acid bacteria (NSLAB) show a different growth trend in cheese, where they are able to reach and maintain high numbers during ripening, without affecting the cheese manufacturing process.

Particular interest has been paid to the use of adjuncts NSLAB for cheese flavor improvement, especially for the manufacture of low- or reduced-fat cheeses that, in most cases, present reduced aromatic properties. An improvement of the overall flavor characteristics has been already documented for reduced-fat Edam cheese [11] and low-fat Caciotta-type cheeses [12].

An example of the interplay between SLAB and NSLAB in dairy microbiota can be found in long-ripened traditional Italian cheeses, such as Parmigiano Reggiano and Grana Padano, that are produced using raw milk and a natural whey starter [13,14]. The use of raw milk leads to the introduction of microorganisms belonging to the *Lactobacillus casei* group; these are initially present in low numbers, but they increase in the subsequent cheese-making steps, and retain viability throughout the ripening time [15]. The *L. casei* group includes three species, *Lactobacillus paracasei*, *Lactobacillus casei* and *Lactobacillus rhamnosus.* Although they are phylogenetically closely related, they present a wide inter- and intra-specific variability [16,17,18]. Recently, Bancalari et al. [19] have proposed to compare the results from impedance analysis and volatile compounds production, to select among strains isolated from raw cow’s milk that show interesting potential as adjunct cultures for the dairy industry.

The application of a combined approach that focused on the relevant outcomes of bacterial fermentation allowed a sensitive and targeted screening of the collections of bacterial isolates for the selection of strains for industrial applications. In this paper, we aimed to find a criterion for selecting strains to use as secondary adjunct starters. To this end, we have evaluated the phenotypic characteristics of the technological interest of a collection of 14 strains belonging to the *L. casei* group, isolated from three different cheeses (Parmigiano Reggiano, Grana Padano and Pecorino Toscano) at different ripening times. In particular, we have evaluated the different attitudes toward the utilization as secondary adjunct cultures of these strains, by assessing their milk acidifying performance through impedance analysis, as well as the production of volatile compounds when grown in milk, using solid-phase microextraction (SPME) and gas chromatography-mass spectrometry (GC-MS). Simultaneously, acetoin metabolic pathway functionality was screened and assessed for its transcriptional activation, to elucidate its activation in response to cultivation in milk.

## 2. Materials and Methods 

### 2.1. Bacterial Strains

Fourteen wild strains belonging to the species *L. casei*, *L. paracasei* and *L. rhamnosus*, isolated from PDO cheeses Parmigiano Reggiano, Grana Padano and Pecorino Toscano at different ripening times were used in this study (Table 1). All the strains belong to the collection of the Laboratory of Food Microbiology of the Department of Food and Drug of University of Parma.

Bacterial strains were maintained as frozen stocks (−80 °C) in Man Rogosa Sharpe (MRS) medium (Oxoid, Milan, Italy) supplemented with 15% glycerol (*w*/*v*). The cultures were propagated three times with a 2% (*v*/*v*) inoculum in MRS and incubated in anaerobiosis (AnaeroGen, Oxoid, Milan, Italy) at 37 °C for 15 h. 

### 2.2. Milk Fermentation and Bacterial Acidifying Performance Analyses 

For milk fermentation studies, strains propagated in MRS broth were used to inoculate 6 mL of full fat ultra-high-temperature (UHT) milk supplemented with 0.1% of yeast extract (Oxoid, Milan, Italy). After an incubation in aerobic condition overnight at 37 °C, each culture was propagated once in 5 mL of unsupplemented full fat UHT milk, and finally inoculated in 30 mL of UHT full cream milk to obtain a final cell concentration of 10^7^ cfu/mL. Of this culture, 10 mL were equally divided in two sterilized GC vials, and incubated at 37 °C for 5 days, for volatile profile analysis. The remaining volume of inoculated milk was equally divided into three previously sterilized measuring vials, and analyzed in triplicate at 37 °C by means of the BacTrac 4300^®^ Microbiological Analyzer system (SY-LAB, Neupurkersdorf, Austria). This instrument consists of two incubators allowing a simultaneous setting of four different temperatures, and was used to evaluate the impedometric curves. M% impedance values were measured and recorded every 20 min for 5 days [20]. One negative sample, consisting of non-inoculated UHT milk, was also measured.

After 5 days of incubation, also the pH was directly measured inside the vials by means of a pHmeter Beckman Instrument mod Φ350 (Furlenton, CA, USA) and glass electrode (Hamilton, Bonaduz, Switzerland). 

### 2.3. Analysis of Volatile Compounds

Volatile compounds produced in fermented milk inoculated with the 14 LAB strains (Table 1), after 5 days at 37 °C, were analyzed by Solid Phase Microextraction-Gas Chromatography-Mass Spectrometry (SPME-GCMS) as described in [19]. Briefly, for each fermented milk produced by strain, volatile compounds analysis in the vial headspace was performed in duplicate using an Agilent Technologies 6890 gas chromatograph coupled to an Agilent Technologies 5873 mass spectrometer (Waters, Milford, MA, USA). 

Before analysis, 5 μL of Toluene (250 mg/L, Sigma, St. Louis, MO, USA) were injected into the sealed vials to be used as an internal standard. An SPME fiber assembly with a divinylbenzene-carboxen-polydimethylsiloxane coating 50/30 μm (Supelco Inc., Bellefonte, PA, USA) was used, and preconditioned according to the manufacturer’s instruction manual. The equilibration step was performed by heating samples at 40 °C for 10 min, followed by an exposition of SPME fiber for 20 min at the same temperature, while applying a magnetic stirring of the vial.

Finally, sample desorption was performed for 3 min at 250 °C. A Zebron ZB-FFAP capillary column (Phenomenex, California, USA, 30 m × 0.25 mm, f.t. 0.25 μm) was used. Volatile compounds separation was performed as follows: Helium carrier gas (1 mL/min), an initial column temperature of 40 °C for 3 min, an increase of temperature to 200 °C at 5 °C/min and maintenance at 200 °C for 5 min. Peak identification from chromatograms was performed by comparison of the experimental mass spectra with those reported in the NIST14 library. A semi-quantitative approach was performed by comparison of the relative peak areas of the volatiles to the peak area of toluene.

### 2.4. Screening for Genes Involved in Acetoin Metabolic Pathway

The 14 LAB strains (Table 1) were screened for the presence of selected genes involved in the synthesis of the volatile compound acetoin. Microbial DNA extraction was performed on overnight cultures using the DNeasy Blood and Tissue Kit (Qiagen, Milan, Italy), following the Gram+ protocol of the manufacturer. DNA was checked for integrity by agarose gel electrophoresis, and purity and concentration was checked spectrophotometrically (NanoDrop™ 2000, Thermo Fisher Scientific, Waltham, MA, USA). Degenerate primers were designed for targeting the following genes: *Ast*, coding for aspartate aminotransferase; *Cly*, coding for citrate lyase, and *Ald*, coding for acetolactate decarboxylase. Details about primer pairs used in this study are reported in Table 2. The 20 µL PCR reaction included 1 µL of genomic DNA, 0.5 µL of 10 µM specific primers, according to the target genes, 10 µL of GoTaq 2× Mastermix (Promega, Milan, Italy) and water up to a volume of 20 µL. The reactions were incubated at 95 °C for 5 min, followed by 30 cycles of 95 °C denaturation for 30 s, annealing at for 30 s and extension at 72 °C for 1 min, and a final extension at 72 °C for 7 min. After the reaction, PCR products were checked on 1.5% agarose gels.

### 2.5. RNA Extraction and cDNA Synthesis

RNA was extracted from UHT full fat milk fermented for 5 days at 37 °C with the strains *L. casei* 2138 and *L. paracasei* 2333, which showed opposite trends for the production of the volatile compound acetoin after growth in UHT full fat milk. RNA was extracted using the TRIzol^®^ Reagent (Invitrogen) protocol. In brief, 250 µL of fermented milk were placed in a 2 mL screw cap tube containing a 500 µL volume of 0.1 mm zirconia/glass beads (BioSpec Products, Bartlesville, USA) and immediately added with 1.5 mL of TRIzol (Invitrogen, Milan, Italy). 

Sample processing was performed with a Mini-BeadBeater 8 (BioSpec Products, Bartlesville, OK, USA), with three repeated intervals of 60 s mixing at maximum speed followed by 60 s pauses on ice. After homogenization, TRIzol manufacturer’s instructions were followed to extract total RNA. Total RNA samples were treated with TURBO DNA-free™ Kit (Ambion, Life Technologies, Milan, Italy). Digested RNA samples were then quantified spectrophotometrically, and 1 µg was used for reverse transcription using SuperScript™ IV Reverse Transcriptase (ThermoFisher Scientific, Milan, Italy) with random hexamers priming strategy, according to the manufacturer’s protocol.

### 2.6. Relative Expression of Ast, Cly and Ald Genes

Real-time quantitative polymerase chain reaction (RT qPCR) was carried out using a QuantStudio^®^ 3 (Thermo Fisher Scientific, Milan, Italy) and the PowerUp SYBR Green Master Mix (Applied Biosystems, Milan, Italy). The 20 µL PCR reaction included 5 µL of 1:10 diluted cDNA, 0.5 µL of 10 µM specific primers (Table 2), 10 µL of 2X Master Mix, and water up to a volume of 20 µL. The reactions were incubated at 95 °C for 10 min, followed by 40 cycles of 95 °C for 15 s and 60 °C for 1 min. After the reaction, the C_t_ data were determined using default threshold settings, the mean C_t_ was determined from three PCR replicates, and calculations were performed using 16S rRNA as a reference gene [21]. The 2^−ΔΔCT^ method was used to determine the relative gene expression, using Pfaffl’s correction [22]. Relative gene expression values are reported as the ratio between the target gene expression level measured in selected strains. The real-time PCR amplification efficiencies (E) in the exponential phase were calculated according to the equation: E = 10 ^(−1/slope)^. The amplification efficiencies of each primer pair are reported in Table 2.

### 2.7. Statistical Analyses

One way analysis of variance (ANOVA) was used to compare the different amounts of volatile compounds produced by the all of the strains, as well as impedance microbiology data, by means of SPSS Statistics 21.0 software (SPSS Inc., Chicago, IL, USA). In this case the Duncan test was adopted to correct for multiple comparisons, and the results were considered different when *p* < 0.05. The concentration values of the volatile compounds detected were analyzed by Principal Component Analysis (PCA) in the R environment (http://www.r-project.org), by using the Factoextra package (v. 1.0.5).

## 3. Results and Discussion

### 3.1. Acidifying Properties of the Strains in Milk

The 14 strains of the *L. casei* group selected for this study were evaluated for their acidifying performance at 37 °C by using the BacTrac 4300^®^ Microbiological Analyzer system. The impedance measurement allows the qualitative and quantitative tracing of the microorganism by measuring the change in the electrical conductivity of the culture medium. Despite the fact that the principle of this technique is not new [23], its application in food microbiology is quite recent, and mainly associated with a rapid detection of foodborne pathogenic bacteria [24].

The most common way to use this measurement was by fixing a point, generally defined as a “time of detection”, that coincides with the reaching of a cell concentration of about 10^6^–10^7^/^mL^ [20]. A new method to use the data recorded has been recently proposed by Bancalari et al. [19] in a work aimed at evaluating the starter LAB acidifying performances. This approach provides three kinetic parameters (Lag, Rate, yEnd) by fitting the impedometric data by the Gomperz equation [20].

The kinetics parameters obtained were used to describe the potential acidifying performances of the tested strains. The results of the analysis of the 14 strains are reported in Table 3 as the mean value of three replicates.

The Lag parameter describes the time that the inoculated cells need to adapt to the growth conditions defined by the analysis. Similarly to the Lag time of a typical bacterial growth curve, the bigger the value, the longer the time that the strains need to adapt to the growth conditions. 

Once adapted, the metabolic activity of the strains modifies the culture media, and thus the measured impedance signal [25]. This parameter was very variable among the strains, ranging from approximately 8 h to 1 h (Table 3). We observed that strain *L. paracasei* 2461 showed the highest Lag time (*p* < 0.05), followed by strains *L. casei* 2138 and *L. rhamnosus* 2233. This behavior can be explained as a reduced adaptability to milk of these three strains. Conversely, the two, *L. rhamnosus* 2075 and 1216, were the fastest to adapt to milk compared with the other strains, showing a Lag value lower than 2 h. The technological meaning of the Rate parameter can be associable to the acidification rate: the higher the value, the faster the acidification rate of the strains. *L. rhamnosus* 2167, together with *L. casei* 2046, showed the faster acidification rate (*p* < 0.05); instead, *L. casei* 1247, *L. paracasei* 2247 and 4201 resulted to be the slower among the tested strains (*p* < 0.05). 

Finally, the last parameter, yEnd, can be used to represent the acidifying capacity of the strains, where the bigger the value, the greater the acidifying capacity. Among all the strains considered, *L. paracasei* 2461 showed the highest value of this parameter (*p* < 0.05), resulting in this the best acidifying strain, while *L. paracasei* 4202 and 2247 showed the lowest values (*p* < 0.05). This feature is of great importance, because the main role of an adjunct culture is to develop during ripening, therefore they must have a long Lag phase to avoid any interference with the activity of SLAB, but they must be also able to survive to the acid environment produced during the first stage of fermentation [4].

### 3.2. Production of Volatile Compounds during Milk Fermentation

A total of 32 volatile compounds were identified in milk fermented for 5 days at 37 °C by the 14 strains of the *L. casei* group, that were assigned to five major classes: acids, alcohols, ketones, esters and other compounds (Appendix A).

Despite the fact that the classes of volatile compounds identified were the same, the total amount of compounds produced is strain-dependent, and some strains were capable to produce higher quantities of compounds with aromatic potential (Figure 1). Strains belonging to the *L. casei* group are taxonomically related, and share similar volatile profiles in terms of the chemical classes of compounds; still, strain-specific differences can be of interest for the selection of secondary adjunct cultures [18,26].

Among the identified volatile compounds, four resulted to vary significantly: 1-hexanol, acetoin, diacetyl and 1-hydroxy-2-propanone (Figure 2). The first compound is an alcohol, that is produced in higher amounts by the strain *L. casei* 2138, that accumulates 8.65 ± 3.96 mg/L (mean ± standard deviation) of 1-hexanol when grown in milk (Figure 2a, *p* < 0.01). This compound has a pleasant fruity/flower aromatic note, and has already been identified in the volatilome of the *L. casei* group isolates [19]. Acetoin and diacetyl are two ketone compounds that are related to the pleasant butter and almonds aroma, and are thus of interest to determine the volatile profile of dairy products [27,28]. The strain able to produce the highest amount of acetoin is *L. casei* 2138 (168.22 ± 47.58 mg/L, Figure 2b), while diacetyl is mainly produced by *L. casei* strain 2186 (10.00 ± 0.79 mg/L, Figure 2c). The high amount of produced diacetyl might derive from the chemical conversion of acetoin, that might be enhanced by a reduced pH in the fermented milk [29]. Finally, 1-hydroxy-2-propanone is a ketone with a pungent and sweet aroma, that is produced up to 2.53 ± 0.94 mg/L from the strain *L. paracasei* 4208 (Figure 2d, *p* < 0.05). This volatile compound has been associated with the volatilome of lactobacilli grown in cheese conditions [26]. It was also reported in the volatile profile of Parmigiano Reggiano cheese, where it represented a small fraction of the identified volatile compounds characteristic of this dairy product [27].

### 3.3. Distribution and Expression of Genes Involved in Acetoin Production 

We observed that the tested strains produce different amounts of acetoin when grown in full-fat milk. The strain *L. casei* 2138 showed the highest production of acetoin, whereas strain *L. paracasei* 2333 is the minor producer after five days of growth. It is known that strains belonging to the *L. casei* group, such as *L. rhamnosus*, encode the metabolic pathway for acetoin production using citrate as a precursor [30]. Other Lactobacillales are capable of converting aspartate into oxaloacetate through a transamination step, that is catalyzed by an aspartate aminotransferase, as described in *Lactococcus lactis* [29,31]. To investigate their metabolic potential, the strains used in this study were screened for the presence of three genes involved in the acetoin pathway: the gene coding for α-acetolactate decarboxylase (*ald*), which converts α-acetolactate into acetoin, and the genes coding for aspartate aminotransferase (*ast*) and citrate lyase (*cly*), which convert respectively aspartate and citrate into oxaloacetate (Figure 3).

Publicly available sequences of the selected genes from 16 *L. casei* group strains (Appendix A) were aligned to design degenerate primers, that were used to screen the fourteen wild isolates. All of the strains showed positive results for the three investigated genes (Appendix A), indicating that they possess the metabolic machinery for the production of acetoin. 

Strains *L. casei* 2138 and *L. paracasei* 2333, that showed opposite acetoin production trends, were tested for the expression of the three genes involved in the acetoin production pathway. The results in Table 4 show that *L. casei* 2138, the major acetoin producer, upregulates the *ast* gene 1.74 folds compared to *L. paracasei* 2333, while the expression of *cly* is downregulated, suggesting a higher activity of the aspartate transaminase in the first strain. Regarding *ald*, coding for the last enzyme in the acetoin pathway, its expression ratio is only slightly upregulated in *L. casei* 2138 compared to *L. paracasei* 2333. These results suggest that the higher amount of acetoin produced by *L. casei* 2138 may be due to a higher activity of the aspartate transaminase pathway, which causes a higher availability of the precursor molecule oxaloacetate. Interestingly, the two strains differ in their measured impedometric parameter Lag, that was of 2.37 h for *L. paracasei* 2333 and 7.1 h for strain *L. casei* 2138. 

According to these results, a high amount of produced acetoin might correlate with the activation of specific aminotransferases (ATs), such as the one encoded by the *ast* gene. ATs activity depends on the availability of α-ketoglutarate, produced by glutamate dehydrogenase (GDH). Different aminotransferases (ATs) could compete for the α-ketoglutarate produced by GDH. As a consequence, the production of aroma compounds depends upon the relative ATs activities towards aspartate or other amino acids of the GDH-positive strain [29]. Another aspect that might prevent acetoin formation via aspartate transaminase is due the fact that not all LAB strains exhibit natural GDH activity [32], and GDH activity among isolates belonging to the *L. casei* group shows strain-specific variability [31,33,34].

Finally, another cause of strain-to-strain differences in acetoin production might be the presence of a frameshift mutation in the acetolactate synthase (*als,*
Figure 3) gene [30]. Anyway, Lo et al. [30] report that the described *als* mutation leads to a complete lack of acetoin/diacetyl synthesis that is not the case for the strain *L. paracasei* 2333. However, other mutations of the *als* coding sequence might affect the activity of the translated enzyme, explaining the observed differences in production rates.

Both the strains, despite the differences observed for acetoin production and transcriptional activation, produce similar amounts of diacetyl (Figure 2c). Conversion of acetolactate into diacetyl is a decarboxylation driven by the redox potential, as shown in previous studies [35,36]. The redox potential of a medium can be modified by charged molecules produced by metabolic activity of the strains, and is therefore linked to their acidifying capacity. The similar diacetyl amount formed in milk cultures by strains *L. paracasei* 2138 and *L. casei* 2333 could be due to their similar acidifying capacity, according to their measured yEnd parameter (Table 3). 

Further studies would be required to assess which of the abovementioned mechanisms might explain the high acetoin production of strain *L. casei* 2138, and the variability observed among strains of the *L. casei* group.

### 3.4. Features-Based Selection of the Strains for Use as Adjunct Cultures

The selection of strains for use as adjunct cultures has to consider the acidifying properties of each strain. Strains with a longer Lag phase are favored in the selection, since they require longer time to adapt to the milk environment, thus avoiding competition in acidification with the starter culture [19]. This is what occurs in the cheese making processes where bacteria belonging to the *L. casei* group are present at low abundances in the vat milk, and remain basically unchanged in number in the first stages of the manufacturing, allowing the natural whey starter to acidify the curd [13,14,15]. For this reason, impedance analysis can provide useful criteria to screen among a wide set of potentially interesting strains [20,25].

In this work, the selected isolates showed differences in the impedometric parameters that are not species nor isolation source related, reflecting the high level of variability often described for the species comprised in the *L. casei* group [17,35]. The Lag parameter was used to discriminate the strains with the major potential as adjunct cultures, selecting those with a longer Lag phase (above five hours).

Figure 4a shows the distribution of the strains according to the three measured parameters: Lag, Rate and yEnd. The strains that show positive values for dimension 1 are characterized by the lowest values for the Rate and Lag parameters, showing that they quickly adapt and grow in the milk environment. Due to their features, these strains would not be ideal as an adjunct, since they might compete with the starter culture. On the other hand, strains clustering in the lower left quadrant of the graph present the highest yEnd values, which means that they possess the best acidifying capacity. This feature might reflect a higher capacity of the bacteria to withstand acid stress, but is not a relevant characteristic for and of the adjunct strain. Strains clustering in the upper left quadrant of the PCA graph are characterized by higher rate values: *L. rhamnosus* 2167 and *L. casei* 2046 are characterized by the highest rate values, and present intermediate Lag values, around three hours, that makes them potentially unsuitable for the intended application. Conversely, the strains *L. paracasei* 4208 and 2247, *L. rhamnosus* 2233 and *L. casei* 2138 present rate values of 1.2 ± 0.02 (mean ± SD), and Lag phases longer than five hours. 

These characteristics make them the most suitable for a potential use as adjunct starter. Furthermore, the four strains present different volatile profiles, as shown in the principal component analysis presented in Figure 4b. Indeed, the strain *L. paracasei* 4208 volatile profile is characterized by a high production of 3-methyl-3buten-1-ol, 2 propanone and 1-hydroxy-2-propanone. Strain *L. paracasei* 2247 is characterized by the highest production of almost all identified compounds, resulting in the most performing aromatic strain. Similarly, strain *L. casei* 2138 produced elevate amounts of volatile compounds, and is the highest acetoin producer among all of the selected strains. Notably, this strain is characterized also by the production of the highest amounts of hexanol and ethanol, which are interesting, since alcohols are the precursors in ester formation [37]. Conversely, strain *L. rhamnosus* 2233 shows among the lowest produced amounts of all of the identified volatile compounds. Still, this strain has interesting acidifying performances, and its suitability as an adjunct culture should be evaluated in a model similar to the desired application, which might not be fermented milk.

## 4. Conclusions

Selection of strains for industrial application needs to be carefully tailored according to the intended fermentation process. Knowledge of the acidifying or aromatic features of the strain is often required to make appropriate choices. In this study, we have identified four strains, belonging to the species *L. casei*, *L. paracasei* and *L. rhamnosus* among the fourteen screened, that could be potentially used as adjunct cultures for cheese-making processes. The combined approach used in this study shows that both the acidifying performance and the production of volatile compounds need to be taken into account for the selection of the best performing strains. This approach has revealed the existence of a wide variability in terms of the protechnological properties of the dairy isolates of the *L. casei* group, even among strains that present similar acidifying properties. Interestingly, the strain that shows the highest production of acetoin appears to upregulate the aspartate pathway, leading to the synthesis of higher amounts of this volatile compound. The observed variability in terms of volatile compounds production after growth in full fat milk opens the perspective of creating bacterial collections that include this information for a more accurate selection of strains for the dairy industry. 

## Figures and Tables

**Figure 1 microorganisms-08-00128-f001:**
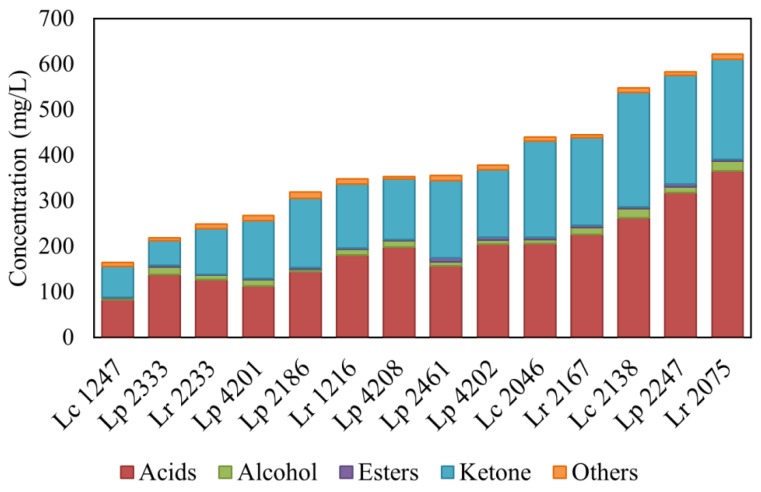
Total volatile compound production for each strain. Strain species are abbreviated as follows: Lc, *L. casei*; Lr, *L. rhamnosus*; Lp: *L. paracasei*. The graph represents the concentration of classes of aromatic compounds produced during growth in full fat milk.

**Figure 2 microorganisms-08-00128-f002:**
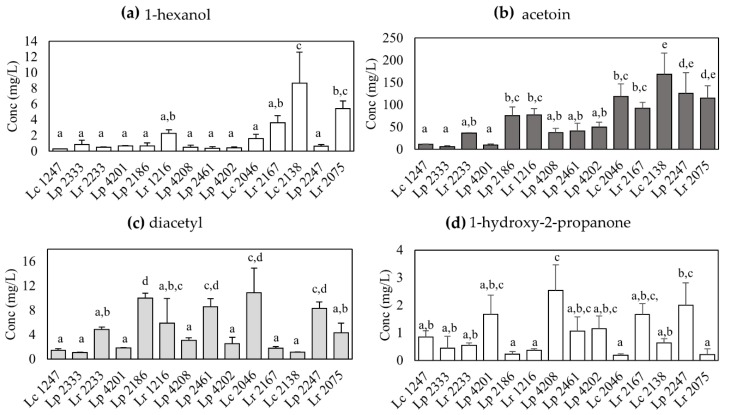
Concentration of statistically significant volatile compounds produced from strains in full fat UHT milk. (**a**) 1-hexanol; (**b**) acetoin; (**c**) diacetyl; (**d**) 1-hydroxy-2-propanone. Strain species are abbreviated as follows: *Lc*, *L. casei*; *Lr, L. rhamnosus*; *Lp*: *L. paracasei.* Concentrations are reported as mg/L. Error bars represent the standard deviation. Different letters in each graph indicate the presence of significant differences according to ANOVA (*p* < 0.05).

**Figure 3 microorganisms-08-00128-f003:**
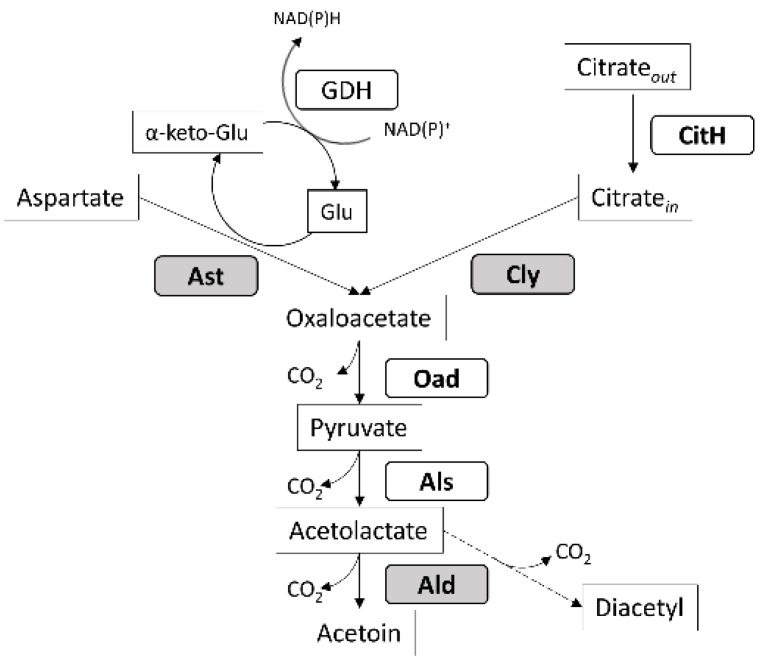
Schematic representation of the acetoin synthesis pathway in *L. casei* group. Ast: Aspartate aminotransferase, Cly: citrate lyase, Oad: Oxaloacetate decarboxylase, Als: Acetolactate synthetase; Ald: Acetolactate dehydrogenase; GDH: Glutamate dehydrogenase; CitH: Citrate permease.

**Figure 4 microorganisms-08-00128-f004:**
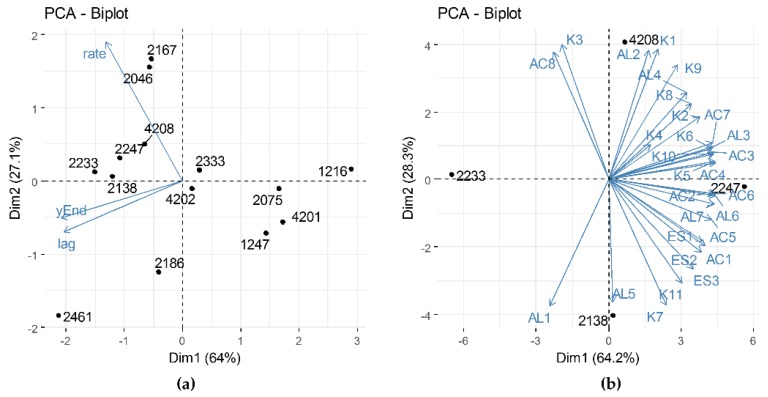
Principal component analysis (PCA) biplot of (**a**) impedometric parameters for each strain and (**b**) of the identified volatile compounds in the selected strains. Position of each strain is reported as a black dot, blue arrows represent variables. For panel (**b**), variable abbreviations are as reported in Appendix A.

**Table 1 microorganisms-08-00128-t001:** Bacterial strains used in this study.

Strain ID	Species	Cheese	Cheese Ripening Time (Months)
1247	*L. casei*	Parmigiano Reggiano	12
2138	*L. casei*	Grana Padano	6
2046	*L. casei*	Grana Padano	2
2247	*L. paracasei*	Parmigiano Reggiano	1
2333	*L. paracasei*	Parmigiano Reggiano	6
2186	*L. paracasei*	Grana Padano	9
2461	*L. paracasei*	Grana Padano	2
4201	*L. paracasei*	Pecorino toscano	2
4202	*L. paracasei*	Pecorino toscano	2
4208	*L. paracasei*	Pecorino toscano	2
2233	*L. rhamnosus*	Parmigiano Reggiano	1
1216	*L. rhamnosus*	Parmigiano Reggiano	12
2167	*L. rhamnosus*	Grana Padano	9
2075	*L. rhamnosus*	Grana Padano	2

**Table 2 microorganisms-08-00128-t002:** Primer pairs used in this study.

Target Name	Primer	Sequence (5′-3′) ^1^	Tm (°C)	PCR Product Size (bp)	Efficiency (E)
alpha-acetolactate decarboxylase (*ald*)	Ald_F	TTCGAAGCCAAGAATATGSCC	55	156	0.93
	Ald_R	GCYTCRGGTCAAAAAGAATT			
aspartate aminotransferase (*ast*)	Ast_F	CAATAACTRATMCGCGCATG	50	143	1.13
	Ast_R	TTCCCGAAYATYAAGCG			
citrate lyase (subunit alpha) (*cly*)	Cly_F	GACAGGTTYTTGATYCCCATT	55	148	1.22
	Cly_R	AACTMGTGGCGTCAATTCA			
16S rRNA	TBA_FW ^2^	CGGCAACGAGCGCAACCC	60	130	0.99
	TBA_RV ^2^	CCATTGTAGCACGTGTGTAGCC			

**^1^** Degenerated bases, reported as indicated in IUPAC base coding, are underlined. ^2^ [21].

**Table 3 microorganisms-08-00128-t003:** Mean values of the three acidifying parameters for each strain. All the results are expressed as mean ± standard deviation (*n* = 3). Different lowercase letters by each column indicate the presence of significant differences according to the analysis of variance (ANOVA) (*p* < 0.05).

Species	Origin	Strain	Lag	Rate	yEnd
***L. casei***	Parmigiano Reggiano cheese	1247	3.43 ± 0.32 ^e^	0.76 ± 0.02 ^a^	28.09 ± 0.56 ^b,c^
***L. casei***	Grana Padano cheese	2046	3.18 ± 0.26 ^d,e^	1.43 ± 0.01 ^e^	29.51 ± 0.10 ^c,d^
***L. casei***	Grana Padano cheese	2138	7.11 ± 0.04 ^g^	1.19 ± 0.01 ^d^	29.30 ± 0.30 ^c,d^
***L. paracasei***	Grana Padano cheese	2186	3.00 ± 0.13 ^d,e^	0.80 ± 0.05 ^b^	30.37 ± 0.38 ^c,d^
***L. paracasei***	Parmigiano Reggiano cheese	2247	6.09 ± 0.17 ^a^	0.61 ± 0.01^a^	23.01 ± 0.17 ^a^
***L. paracasei***	Parmigiano Reggiano cheese	2333	2.38 ± 0.15 ^c,d^	1.03 ± 0.00 ^c^	29.62 ± 0.02 ^c,d^
***L. paracasei***	Grana Padano cheese	2461	8.63 ± 0.74 ^h^	0.86 ± 0.03 ^b^	30.91 ± 0.06 ^d^
***L. paracasei***	Pecorino Toscano cheese	4201	2.12 ± 0.38 ^c^	0.75 ± 0.02 ^a^	28.24 ± 0.44 ^b,c^
***L. paracasei***	Pecorino Toscano cheese	4202	4.78 ± 0.34 ^f^	0.84 ± 0.01 ^b^	21.70 ± 0.30 ^a^
***L. paracasei***	Pecorino Toscano cheese	4208	5.31 ± 0.88 ^f^	1.22 ± 0.01 ^d^	29.19 ± 0.82 ^c,d^
***L. rhamnosus***	Parmigiano Reggiano cheese	1216	1.20 ± 0.07 ^a,b^	0.80 ± 0.00 ^b^	26.69 ± 0.42 ^b^
***L. rhamnosus***	Grana Padano cheese	2075	1.98 ± 0.12 ^b,c^	0.86 ± 0.01 ^b^	28.14 ± 0.18 ^b,c^
***L. rhamnosus***	Grana Padano cheese	2167	3.61 ± 0.27 ^e^	1.45 ± 0.01 ^e^	29.17 ± 0.28 ^c,d^
***L. rhamnosus***	Parmigiano Reggiano cheese	2233	6.69 ± 0.25 ^g^	1.22 ± 0.01 ^d^	29.88 ± 0.06 ^c,d^

**Table 4 microorganisms-08-00128-t004:** Relative gene expression of the selected genes in the *L. paracasei* strains in response to growth in full fat milk. The data are reported as the ratio between the target gene expression level in the high acetoin producer strain *L. casei* 2138. strain and *L. paracasei* 2333.

Target	Strains	Ratio	Ratio SD	Efficiency
*ald*	2138	1.108	0.012	0.93
2333			
*ast*	2138	1.743	0.015	1.13
2333			
*cly*	2138	0.324	0.014	1.22
2333			
*16S rRNA*	2138	1.000	0.012	0.99
2333

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
