# Peer review of "Phenotypic Diversity of Lactobacillus casei Group Isolates as a Selection Criterion for Use as Secondary Adjunct Starters"

_microorganisms, 2020, doi:10.3390/microorganisms8010128_

Round 1

Reviewer 1 Report

Corrections needed:

In Table 2 degenerated bases are not shown (not underlined) - please indicate the variable bases; Table 3 – use dot instead of comma! Discussion of gene expression part of study needs explanation why authors state on lines 292-297: „Finally, another possibility for strain-to-strain differences in acetoin production, might be the presence of a frameshift mutation in the acetolactate synthase (als, Figure 3) gene [30]. Anyway, the authors of [30] report that the described als mutation leads to a complete lack of acetoin/diacetyl synthesis, that is not the case for strain L. paracasei 2333. Further studies would be required to assess which of the abovementioned mechanisms might explain the high acetoin production feature of strain L. casei 2138.“         It is shown in authors own data in Supplementary table 1 about diacetyl content (K4), that in diacetyl content in samples treated by abovementioned strains is equal (1.1 mg/L). Both acetoin and diacetyl are directly derived from acetolactate (Fig 3).  If the synthesis of acetolactate is affected by any reason, this should lead to lower formation of diacetate also. Please revise this part of discussion or omit it.

Author Response

Corrections needed:

In Table 2 degenerated bases are not shown (not underlined) - please indicate the variable bases;

Degenerated bases are now correctly indicated.

Table 3 – use dot instead of comma!

The Table was corrected.

Discussion of gene expression part of study needs explanation why authors state on lines 292-297: „Finally, another possibility for strain-to-strain differences in acetoin production, might be the presence of a frameshift mutation in the acetolactate synthase (als, Figure 3) gene [30]. Anyway, the authors of [30] report that the described als mutation leads to a complete lack of acetoin/diacetyl synthesis, that is not the case for strain L. paracasei 2333. Further studies would be required to assess which of the abovementioned mechanisms might explain the high acetoin production feature of strain L. casei 2138.“ 

The sentence was rephrased as follows:

Anyway, the authors of [30] report that the described als mutation leads to a complete lack of acetoin/diacetyl synthesis, that is not the case for strain L. paracasei 2333. However, other mutations of als coding sequence might affect the activity of the translated enzyme, explaining the observed differences in production rates.

It is shown in authors own data in Supplementary table 1 about diacetyl content (K4), that in diacetyl content in samples treated by abovementioned strains is equal (1.1 mg/L). Both acetoin and diacetyl are directly derived from acetolactate (Fig 3).  If the synthesis of acetolactate is affected by any reason, this should lead to lower formation of diacetate also. Please revise this part of discussion or omit it.

We thank the reviewer for the useful comment. To improve the discussion, we added the following sentence in the text (line 350):

Both the strains, despite the differences observed for acetoin production and transcriptional activation, produce similar amounts of diacetyl (Figure 2c). Conversion of acetolactate into diacetyl is a decarboxylation driven by the redox potential, as shown in previous studies [35, 36]. The redox potential of a media can be modified by charged molecules produced by metabolic activity of the strains, and is therefore linked to their acidifying capacity. The similar diacetyl amount formed in milk cultures by strains L. paracasei 2138 and L. casei 2333 could be due to their similar acidifying capacity, according to their measured yEnd parameter (Table 3).

Reviewer 2 Report

The authors present a simple and elegant study that investigates the phenotypes of L. casei strains used in cheese fermentation. This study should be of moderate interest to the cheese producing industry. However, in this reviewer’s opinion, this manuscript needs some improvement before publication should be considered. The English language should be thoroughly edited and improved. There are several sentence structure problems as well as other paragraphs that are hard to understand. I tried to point out some of the issues in my minor comments but I gave up after a while. I know that MDPI provides editing services. I suggest investing in that.

The experimental procedure is generally sound except for some issues that I point out in my major comments.

Major comments:

For results shown in Table 4 when was the measurement taken? Beginning of Log phase, middle, end? According to line 141 the RNA was only extracted 5 days after culture inoculation. Is this the case? If so, is there evidence that action production occurs in the stationary phase which I assume is the phase that these strains reached after 5 days? If not, in this reviewer’s opinion, the conclusion about gene expression and action production is not founded. Related to the point above the authors state that the two strains differ in the length of the lag phase. The lag phase is usually the phase when the bacteria ramp up gene expression to metabolize the food in the media. Are you suggesting that he longer lag phase for L. casei is because they upregulate gene expression for acetoin production? If this is the case, gene expression should be measured right after the end of the lag phase. Another parameter that the authors don’t take into consideration is the Km (affinity) and Kcat (turnover) parameters of these enzymes. The authors conclude that a 1.7X upregulation is indicative of ramping up the oxaloacetate to aspartate pathway but a 2X higher affinity of one enzyme could easily make up for the decreased amount of molecules. E.g. you need more of the slower enzyme to produce the same amount of product as a faster enzyme. The authors should research these parameters for the enzymes and include it in their interpretation.

Minor comments:

Lane 13: “characterized…characteristics.” Try using a synonym. Line 20: “upregulated the aspartate pathway.” Line 21: “An increased knowledge of the…” Line 27: “Before starter cultures introduction in the 19th century…” No need for a comma after “introduction.” Line 29. Delete “also.” Line 40: Sentence starting “With this aim…” is awkward. Consider re-writing. Line 46: What is meant by “opposite kinetic of growth”? To this reviewer this doesn’t sound like the right terminology for microorganisms physiology. Please re-write. Line 51: What is meant by “an improvement of the sensory attributes”? Are you referring to taste or to bacteria environmental (quorum) sensing? Line 66: Consider splitting the sentence into two. “In this paper we aimed to find a …adjunct starters.” “To this end, we have evaluated…” Line 100. “After five days of incubation, the pH was measured…” Please give PCR primer concentration. The volume is meaningless without the concentration. Line 169: “One way ANOVA was used…” Line 179: “Despite the fact that the…” Line 181: “…this measurement was by fixing…” Line 191: …”that the strains need to modify…” What do you mean by this? Do you mean adaptation to the culture by ramping up transcription of relevant genes? Also line 192: I assume you mean “phase” not “fase.” For Table 2 I suggest using the notation 3.43+/-0.43 instead of commas. Line 226: “…only four resulted to vary significantly” is awkward. Consider rewriting.

Author Response

Comments and Suggestions for Authors

The authors present a simple and elegant study that investigates the phenotypes of L. casei strains used in cheese fermentation. This study should be of moderate interest to the cheese producing industry. However, in this reviewer’s opinion, this manuscript needs some improvement before publication should be considered. The English language should be thoroughly edited and improved. There are several sentence structure problems as well as other paragraphs that are hard to understand. I tried to point out some of the issues in my minor comments but I gave up after a while. I know that MDPI provides editing services. I suggest investing in that.

The experimental procedure is generally sound except for some issues that I point out in my major comments.

We thank the reviewer for the precious comments that helped to improve the quality of the manuscript. As suggested, the English has been reviewed, and all the changes included in the reviewed version of the manuscript.

Major comments:

For results shown in Table 4 when was the measurement taken? Beginning of Log phase, middle, end? According to line 141 the RNA was only extracted 5 days after culture inoculation. Is this the case? If so, is there evidence that action production occurs in the stationary phase which I assume is the phase that these strains reached after 5 days? If not, in this reviewer’s opinion, the conclusion about gene expression and action production is not founded.

We thank the Reviewer for the comment. Measurement of RNA expression was performed after 5 days incubation in milk. A recent work performed on microorganisms of the L. casei group grown in elderberry juice (Ricci et al., 2018) has shown that also bacterial cells that retain a non growing state are capable to synthesize acetoin and diacetyl, even in large amounts. Similarly, Diaz-Muniz et al. (2005) observe that L. casei ATCC334 cells grown in a cheese-mimicking environment accumulate acetoin and diacetyl during the late exponential and stationary phase. Other studies do not emphasize the growth kinetics of bacterial cells, but these evidences might suggest that accumulation of acetoin and diacetyl during the late exponential/stationary phase can be observed in LAB grown on various substrates.

Related to the point above the authors state that the two strains differ in the length of the lag phase. The lag phase is usually the phase when the bacteria ramp up gene expression to metabolize the food in the media. Are you suggesting that he longer lag phase for L. casei is because they upregulate gene expression for acetoin production? If this is the case, gene expression should be measured right after the end of the lag phase.

We thank the reviewer for the observation, and of course it would have been interesting to measure gene expression after the end of lag phase. Still, according to the aim of this study, measurement of the Lag parameter, that is an “impedance parameter” and not the duration of lag phase (Bancalari et al 2016, Bancalari et al 2019), is used to select strain that perform well as adjunct starters, and we do not know if upregulation of genes involved in acetoin production occurs during this phase. Otherwise, this aspect is of outmost interest, yet far from the aim of our research, requiring targeted and in-depth research that could be addressed in the future.

Another parameter that the authors don’t take into consideration is the K(affinity) and Kcat (turnover) parameters of these enzymes. The authors conclude that a 1.7X upregulation is indicative of ramping up the oxaloacetate to aspartate pathway but a 2X higher affinity of one enzyme could easily make up for the decreased amount of molecules. E.g. you need more of the slower enzyme to produce the same amount of product as a faster enzyme. The authors should research these parameters for the enzymes and include it in their interpretation.

We agree with the reviewer that upregulation of gene expression cannot provide, alone, a complete explanation of the observed phenotypic diversity in acetoin formation from bacteria belonging to the L. casei group. However, to our knowledge, no studies have investigated the kinetic parameters of Aspartate transaminase and Citrate lyase from L. casei group, thus no conclusive observations can be drawn about this. However, in the present study, we just report that there is an increased expression of the Ast encoding gene in strain L. paracasei 2138. As for the question above, further studies are needed to understand if this overexpression actually leads to an increased presence of Ast protein and if this can explain the higher acetoin production rates. However, in order to answer these questions, targeted research is needed which could be addressed in future works.

Minor comments:

Lane 13: “characterized…characteristics.” Try using a synonym.

The text was modified accordingly

Line 20: “upregulated theaspartate pathway.”

The text was modified accordingly

Line 21: “An increased knowledge of the…”

The text was modified accordingly

Line 27: “Before starter cultures introduction in the 19th century…” No need for a comma after “introduction.”

The text was modified accordingly

Line 29. Delete “also.”

The text was modified accordingly

Line 40: Sentence starting “With this aim…” is awkward. Consider re-writing.

The text was modified

Line 46: What is meant by “opposite kinetic of growth”? To this reviewer this doesn’t sound like the right terminology for microorganisms physiology. Please re-write.

The sentence was re-phrased as follows:

“non starter lactic acid bacteria (NSLAB) show a different growth trend in cheese, where they are able to reach and maintain high numbers during ripening without affecting the cheese manufacturing process.”

Line 51: What is meant by “an improvement of the sensory attributes”? Are you referring to taste or to bacteria environmental (quorum) sensing?

The sentence was re-phrased as follows:

An improvement of the overall flavor characteristics has been already documented for reduced-fat Edam cheese [11], and low-fat Caciotta-type cheeses [12].

Line 66: Consider splitting the sentence into two. “In this paper we aimed to find a …adjunct starters.” “To this end, we have evaluated…”

The aim was divided in two sentences, as suggested.

Line 100. “After five days of incubation, the pH was measured…”

The text was modified accordingly

Please give PCR primer concentration. The volume is meaningless without the concentration.

Primer concentration was added at line 136

Line 169: “One way ANOVA was used…”

The sentence was rephrased accordingly

Line 179: “Despite the fact that the…”

The text was modified

Line 181: “…this measurement was by fixing…”

The text was modified

Line 191: …”that the strains need to modify…” What do you mean by this? Do you mean adaptation to the culture by ramping up transcription of relevant genes?

Yes, indeed during Lag phase adaptation of the strains to the culture occurs, due to regulation of transcription, that subsequently leads to modulation of the metabolic activity. The sentence was rephrased to improve clarity:

“…the longer the time that the strains need to adapt to the growth conditions. Once adapted, the metabolic activity of the strains modifies the culture media and thus the measured impedance signal ”

Also line 192: I assume you mean “phase” not “fase.”

We have corrected the typing error

For Table 2 I suggest using the notation 3.43+/-0.43 instead of commas.

The text and table were modified accordingly

Line 226: “…only four resulted to vary significantly” is awkward. Consider rewriting. 

The sentence was rewrited as follows:

“Among the identified volatile compounds four resulted to vary significantly: 1-hexanol, acetoin, diacetyl and 1-hydroxy-2-propanone (Figure 2).”